# Object-Cooperated Ternary Tree Partitioning Decision Method for Versatile Video Coding

**DOI:** 10.3390/s22176328

**Published:** 2022-08-23

**Authors:** Sujin Lee, Sang-hyo Park, Dongsan Jun

**Affiliations:** 1School of Computer Science and Engineering, Kyungpook National University, Daegu 41566, Korea; 2Department of Computer Engineering, Dong-A University, Busan 49315, Korea

**Keywords:** ternary tree, versatile video coding, object detection, encoding complexity, machine learning

## Abstract

In this paper, we propose an object-cooperated decision method for efficient ternary tree (TT) partitioning that reduces the encoding complexity of versatile video coding (VVC). In most previous studies, the VVC complexity was reduced using decision schemes based on the encoding context, which do not apply object detecion models. We assume that high-level objects are important for deciding whether complex TT partitioning is required because they can provide hints on the characteristics of a video. Herein, we apply an object detection model that discovers and extracts the high-level object features—the number and ratio of objects from frames in a video sequence. Using the extracted features, we propose machine learning (ML)-based classifiers for each TT-split direction to efficiently reduce the encoding complexity of VVC and decide whether the TT-split process can be skipped in the vertical or horizontal direction. The TT-split decision of classifiers is formulated as a binary classification problem. Experimental results show that the proposed method more effectively decreases the encoding complexity of VVC than a state-of-the-art model based on ML.

## 1. Introduction

Deep learning (DL) [1] techniques are known to outperform non-DL approaches in diverse fields, such as computer vision [2], natural language processing [3], and speech recognition [4]. Among the DL models, object detection models have demonstrated particular success in computer vision but have rarely been deployed in video coding. Instead, most researchers who have investigated video coding use traditional machine learning (ML) approaches to reduce the complexity of the encoders [5]. Herein, we propose that object detection models can reduce the video coding complexity of the next-generation video coding standard known as versatile video coding (VVC) [6].

Unlike previous video coding standards, such as high efficiency video coding (HEVC) [7], VVC introduces a multi-type tree (MTT) block partitioning structure that supports binary tree (BT) and ternary tree (TT) splits in the horizontal and vertical directions. Although these approaches achieve higher coding efficiency than HEVC, they considerably increase the computational complexity [8] because the VVC encoder applies a brute-force method to optimize the partitioning structure.

To reduce the encoding complexity, we must reduce the number of MTT-partitioning steps. Recent complexity reduction methods have achieved fast MTT-partitioning decision in VVC [5]; however, the methods that use object detection for fast TT partitioning in the horizontal and vertical directions are rarely reported.

To bridge this gap, we previously proposed improving the complexity of the VVC encoder by applying ML models. This research is an extension of work originally presented in ICEIC 2022 [9]. In this paper, we first apply object detection techniques to VVC to decrease its encoding complexity. We propose a new framework by combining DL for object detection in the first stage (feature extraction) and ML for accurate TT-split prediction in the second stage (TT-split decision). In the feature extraction stage, we extract new features related to objects via object detection in each frame of a video sequence while obtaining the context-based features using the traditional context-based approach. In the TT-split decision stage, we conduct experiments using ML models with extracted features for fast TT partitioning to reduce the encoding complexity of VVC.

Herein, we show that our proposed method decreases the encoding time of the VVC test model (VTM4.0) by up to 60% with an average coding loss of 0.56%. Moreover, our proposed method is flexible and adaptable to applications.

The contributions of this paper are summarized as follows:We discover new object-based features that can cooperate with context-based methods. We assume that objects are the key characteristics of videos and object features can help reduce the computational cost (or complexity) of VVC.Our proposed framework newly combines a descent DL-based model with the traditional ML approach—DL for feature extraction and ML for the final decision scheme. Experimental results show that the proposed combined framework outperforms the state-of-the-art model.

The remainder of this paper is organized as follows. Section 2 describes existing methods related to encoding complexity reduction and overviews DL models for object detection. Section 3 introduces the proposed object-cooperated TT partitioning decision method. The dataset and the associated training process are also presented in Section 3. Section 4 describes the experimental setup and test environment and compares the performances of the MLP-based method and proposed methods with that of the anchor. Finally, Section 5 provides conclusions of this study.

## 2. Related Works

### 2.1. Existing Methods for Encoder-Complexity Reduction

Although VVC can provide powerful coding performance, the computational complexity of the VVC encoder is substantially high compared with that of the HEVC encoder. The current various approaches apply statistical analysis and neural networks have been researched to reduce the complexity of the VVC encoder. For a statistical analysis based approach, Park et al. [8] proposed a context-based fast TT decision method using the directional information between BT and TT. H. Yang et al. [10] proposed a fast intra coding algorithm consisting of fast coding unit (CU) partition and fast intra mode decision using the combination of binary classifiers. This method exploited the textural complexity of the current CU and the context information from neighboring CUs.

The aforementioned methods evaluated the statistical correlations between a current CU and the neighboring CUs. Recent studies have studied neural network-based fast decision schemes based on neural networks to avoid redundancy in the process of optimal VVC block structure. Park et al. [11] designed a fast decision scheme using two lightweight neural networks to determine TT block partitioning. Q. Zhang et al. [12] proposed a fast CU decision algorithm based on DenseNet, which predicts the probabilities of whether the edge of 4 × 4 blocks are the boundary blocks or not. T. Li et al. [13] designed a multi-stage Convolutional Neural Network (CNN) model to predict the quad-tree and multi-type tree-based CU partition method for accelerating the encoding process of intra-mode VVC. S. Wu et al. [14] proposed a hierarchy grid fully convolutional network framework, which can substantially predict the particular hierarchical split structure to automatically control the trade-off between coding efficiency and complexity.

### 2.2. Object Detection

Object detection employs computer vision and image processing technologies to detect object instances of a certain class within an image. Object detection can be categorized in: ML-based and DL-based approaches. ML-based approaches are frameworks based on Haar features, scale-invariant feature transform (SIFT), and histograms of oriented gradients (HOG) functions [15] followed by a classification technique such as a support vector machine (SVM). Meanwhile, DL techniques perform end-to-end framework without using specifically defined functions and are usually based on CNN. DL-based methods for object detection can be categorized into two main types: one-stage methods and two-stage methods. The one-stage methods prioritize the inference speed using You Only Look Once (YOLO) [16], a single shot detector [17], or RetinaNet [18]. The two-stage methods prioritize the detection accuracy using a model such as Faster R-CNN [19], Mask R-CNN [20], or Cascade R-CNN [21].

In this study, we use a model that prioritizes the inference speed to reduce the encoding complexity of VVC. Therefore, we choose the YOLO method for object detection and the YOLOv5 [22] model for experiments. YOLOv5 has lower capacity and faster speed than the other YOLO versions. YOLOv5, a family of object detection architecture and models pre-trained on the COCO 2017 dataset [23], has been introduced to four models: YOLOv5s, YOLOv5m, YOLOv5l, and YOLOv5x, which are simply named small, medium, large, and extra-large, respectively. The four models have the same backbone or head, but different multiples of model depths and layer widths. We use the YOLOv5s model, which has the fastest inference speed among the YOLOv5 pre-trained models.

## 3. Proposed Object-Cooperated TT Partitioning Decision Method

Because the VVC encoder cannot know which block should avoid the TT-split process [9], it usually attempts to determine the optimal partitioning structure via brute-force search, which is a time-consuming process. To reduce the encoding time from that of brute-force searching, we adopt ML models that decide when a TT-split is required. First, using ML models, we experimentally identify which model can accurately predict whether a TT-split process should be skipped (see Section 4.3). Second, we select the lightweight model that provides the highest accuracy of TT-split decisions with our extracted features compared with the existing method (see Section 4.4). In our proposed method, the model that makes accurate TT-partitioning decisions using extracted features is applied to each TT-split direction.

As mentioned above, our proposed method for fast TT partitioning comprises two-stages: the feature extraction stage and the TT-split decision stage. Figure 1 presents the framework of the proposed object-cooperated TT partitioning decision method. In the feature extraction stage, our extracted features related to objects are obtained via DL-based object detection and the context-based features [11] are obtained from the traditional context-based approach for each TT-split direction. The feature extraction and the training and evaluation datasets are explained in Section 3.1. In the TT-split decision stage, two decision tree (DT) classifiers are applied in each TT-split direction in the same manner as done in [11]: horizontal TT (TT_H) and vertical TT (TT_V). The encoding-complexity reduction of VVC by the DT model using the extracted features is experimentally demonstrated in each TT-partitioning direction.

### 3.1. Process of Feature Extraction including Object Detection

To reduce the encoding complexity of VVC, the extracted features forming the input vector of the ML model must ensure accurate predictions of TT-splits decisions using the model. Accordingly, we assume that objects are among the main features of the video sequences; moreover, we can characterize the number and ratio of objects. As shown in Figure 1, our feature extraction method uses YOLOv5 object detection. We first extracted 11 features such as quad-tree depth, MTT depth, etc. (Features *F* presented in Figure 1) through context-based approaches. Second, we executed YOLOv5 based on object detection on the frames of video sequences to obtain object-cooperated features (Features *O* presented in Figure 1). Figure 2 shows the object detection results of two frames in video sequences. In the first frame (Figure 2a), 23 objects were detected: 9 persons, 12 bicycles, and 2 backpacks. We define the object’s ratio as the ratio of objects, and the objects’ number as the sum of the number of objects by object detection in the frame. The object’s number is 23 and the object’s ratio is 51.58%. The object’s number and objects’ ratio are the new features obtained from object detection. We specified the new features as object features. By contrast, no objects were detected in the second frame (Figure 2b). In this case, the object’s number and objects’ ratio are zero. Object’s ratio is defined as follows: (1)$$\mathrm{Object}’s\;\mathrm{ratio}=\frac{A\;\mathrm{bounding}\;\mathrm{box}\;\mathrm{size}\;\mathrm{of}\;\mathrm{objects}\;\mathrm{detected}\;\mathrm{by}\;\mathrm{YOLOv}5}{A\;\mathrm{frame}\;\mathrm{resolution}}\times 100,$$

Features *F* were extracted during the encoding process. The datasets obtained from encoding process comprised 11 input features and a binary class—determining whether the TT-split process is required (binary class = 1) or not (binary class = 0). The features *F* were the same as those reported in [11]. The 11 features are the quad-tree depth (QTD), BT’s superiority in rate-distortion (RD) cost view (BTS), Boolean value indicating whether the optimal BT direction of two BTs in RD cost view (BTD), block shape ratio depending on TT direction (BSR), BT/TT depth (MTD), intra prediction mode (IPM), intra subblock partition (ISP), multiple reference lines (MRL), coded block flag (CBF), multiple transform set (MTS), and quantization parameter value for a frame (QP). The reported 11 features details can be found [11]. Features *O* present the object’ number and object’s ratio of a frame in a video sequence. We added two columns of the input vector of the ML models to combine the newly extracted Features *O* with Features *F* obtained using the existing method [11].

### 3.2. Datasets for Training and Evaluation

The training dataset in this study was derived from the Tencent Video Dataset (TVD) [24], which differs from that in [11]. The TVD captures a variety of content coverage within 86 video sequences, each comprising 65 frames with 3840 × 2160 spatial resolution. As the training dataset, the 0th frame (the very first frame of the sequence), the 20th frame, the 40th frame, and the 60th frame of TVD sequences were used and then encoded under an all intra (AI) configuration. The frames used for training were never used for testing. Five QP values with a wide range (20, 25, 30, 35, and 40) were used for encoding.

We used JVET test sequences [25] to evaluate the proposed method. For encoding, we adopted QP values with a wide range of 20, 25, 30, 35, and 40. The test dataset comprised the encoded 0th frame in each test video sequence. Features were extracted during the encoding process. To evaluate the performance of the ML models, object-features in the 0th frame of each test sequence were extracted via object detection. Moreover, when comparing the performances of the existing and proposed complexity-reduction methods, we used object features obtained via object detection in the common test condition (CTC) recommended by JVET experts [25].

### 3.3. Data Augmentation of the Training Datasets

To improve the generalization of the ML model, we expanded the training dataset through a data augmentation technique called pixel-level transform, which was implemented via OpenCV-Python [26]. This technique adjusts the brightness of the sequences by manipulating the pixel values in the original video sequences. As shown in Table 1, 10 video sequences were adjusted to be brighter (by 39.06%) than the original video sequences and another 10 video sequences were adjusted to be darker (by −19.53%) than the original sequences. Figure 3 shows examples of an original sequence, a brighter sequence, and a darker sequence.

Table 2 lists the number of training samples collected after data augmentation of TVD. The original dataset comprises 86 sequences. Each brightness dataset consists of 10 sequences that were brightened and 10 sequences that were darkened via data augmentation. A total of 6,665,015 and 6,689,424 samples were thus collected for TT_H and TT_V, respectively.

To demonstrate the effectiveness of the data augmentation technique, we used the Pearson correlation coefficient (PCC) to analyze the correlation between extracted features and binary classes in Figure 4. Figure 4a,b display the heatmaps of PCCs before and after data augmentation, respectively, for TT_H. One of the object-features obtained via object detection, i.e., object’s ratio increased to 0.0085 and 0.012. It was confirmed that the feature was more related to the binary class. In the next experiment, we applied the DT model as the basic ML model with different maximum tree depth (max depth = 5, 6, and 7) to TT_H splitting decisions. The performances of the models with and without data augmentation are compared in Table 3. Because approximately 10% of datasets were added through the data augmentation technique and accuracy of the DT model increased as the depth of the DT increased, we applied the data augmentation technique to training datasets. In addition, when object detection is performed, it is reported that the effect is better if the data augmentation technique is applied [27]. Thus, the data augmentation technique is applied to the experiment.

### 3.4. TT Partitioning Decision Stage Based on DT

To decide whether TT partitioning should be avoided, we proposed a framework using two DTs for the TT-split decision stage (Figure 1). DTs were chosen owing to their very fast inference speed and low implementation complexity on DTs with limited maximum depth. DT is a nonparametric supervised learning algorithm for classification and regression. The model generated using a DT predicts the value of a target by learning simple decision rules inferred from the data features. To predict the value of a target, we used the Gini impurity function [28], which determines how well a DT is split. The Gini impurity ranges from 0 (all elements belong to the same class) to 1 (each class has only one element). When Gini impurity is 1, all elements are randomly distributed into various classes; when the Gini impurity is 0.5, the elements are uniformly distributed across some classes. Furthermore, a DT learns from the data features and approximates a sine curve with a set of if-then-else decision rules up to max depth. The deeper the tree is, the more complicated the rules of the DT model are. The DT is usually divided until the class value is perfectly determined or until the data are fewer than the minimum number of samples that can be split; that is, the minimum number of sample data required to form a leaf node.

Because TT-splits are directional, DTs are trained separately. Thus, the Gini values resulting from the DTs, which determine whether a TT should be split (i.e., TT partitioning), differ between the models. Among the ML models for determining TT partitioning, DT was chosen because it shows the best accuracy. The performances of the ML models in each direction are given in Table 4 and Table 5 (see Section 4.3).

We established two models for the two TT-split directions (TT_H and TT_V). In each model, we evaluated three DT models with different max depths (5, 6, and 7). The range of max depth was limited for the following reasons:If the max depth is less than 5, the model is oversimplified and provides poor predictions.If the max depth is greater than 7, the model becomes too complicated and is prone to overfitting.

During the experiment, the DT with max depth = 7 achieved the best predictions; therefore, the maximum depth was set to 7 in subsequent analyses. Figure 5 shows graphs of the DT models with max depth = 7. Figure 5a,b are the left side and right-side graphs of the DT based on the root node for TT_H. Figure 5c,d are the left side and right-side graphs of the DT based on the root node for TT_V. The graphs can be enlarged by running our GitHub code (https://github.com/sujineel/Object-cooperated-Ternary-Tree-Partitioning-Decision-Method-for-Versatile-Video-Coding accessed on 12 June 2022), which is provided online.

The output value *y* of the DT model determines whether TT should be split in each direction. For example, if the output value of a DT, *y*, is equal to 1, TT is split; if *y* = 0, TT is not split. We set a threshold α in the mid-range of *y* (i.e., when *y* ranges between 0 and 1, α = 0.5) for mapping the floating value to a Boolean value (true/false answer). The entire TT_H or TT _V splitting process is omitted if *y* is less than 0.5. In the entire TT-partitioning process, *y* determines the best CU. By avoiding unnecessary TT-splits, the proposed method reduces the encoding complexity of VVC.

The coding efficiency is considerably reduced when a required TT is incorrectly predicted by the DT; that is, when the DT outputs a false-negative. To solve the coding loss of the predictive DT model, we should adjust the α to suit the encoding application. If the application prioritizes image quality, α should be less than 0.5 even if the complexity is somewhat compromised. Therefore, we propose two threshold values (0.5 and 0.25) that accomplish a reasonable trade-off between coding efficiency and complexity.

## 4. Experimental Results

### 4.1. Experimental Setup

All the encoding operations were conducted using personal computers with Intel i7-10700 eight-core 2.90-GHz processors and a 64-bit Windows 10 operating system, with the hyper-threading and turbo modes turned off. Experiments were performed without GPUs to reduce the complexity of the ML models. The training and testing of the ML models were assessed using Jupyter Notebook. Visual Studio 2017 was used for conversion and experiments with c++ languages.

The model performances were evaluated using the TensorFlow [29] and scikit-learn libraries [30]. The TensorFlow library is an open-source software library for ML and artificial intelligence. TensorFlow can be used for a range of tasks but focuses particularly on the training and inference of deep neural networks. TensorFlow was developed by the Google Brain team for internal Google use in research and production. Scikit-learn is a free software ML library for the Python programming language. Scikit-learn includes various classification, regression, and clustering algorithms, including SVM, random forest (RF), gradient boosting, and k-means. It is designed to inter-operate with the numerical and scientific Python libraries such as NumPy and SciPy.

### 4.2. Performance Metrics of the Proposed Method

#### 4.2.1. Evaluation Metrics of the ML Models

ML models are used in the TT-split decision stage of the proposed model. As the performance metrics of the ML models, we used the metric provided by TensorFlow and Scikit-learn libraries. We measured total time for training and the accuracy of models. The accuracy is defined as follows: (2)$$Accuracy=\frac{TruePositives+TrueNegatives}{TruePositives+TrueNegatives+FalsePositives+FalseNegatives},$$
where *True Positives* denote the correct predictions of actually true answers, *False Positives* denote the wrong predictions of actually false answers predicted as true, *True Negatives* denote the correct predictions of actually false answers, and *False Negatives* denote the wrong predictions of actually true answer predicted as false.

To evaluate the accuracy of our method using the object-features, we added the object-features to the DT with max depth = 7 according to the direction of TT-splits. The results of the experiment are shown in detail in Section 4.4.

#### 4.2.2. Evaluation Protocol of the Proposed Method: Comparisons with the Anchor

All encoding experiments were conducted using VTM4.0 in the AI coding configuration. To evaluate the performance of the proposed method, coding efficiency and computational complexity were measured in terms of Bjontegaard delta bit rate (BDBR), which represents the rate saving of methods under the same objective quality and computing encoding time (ΔEncT). The BDBR is the bitrate loss over four QPs in percentage with respect to the anchor for the same Peak Signal-to-Noise. In [11], BDBR is defined as follows: (3)$${BDBR}_{yuv}=\frac{\left(6BDBR_{y}+BDBR_{u}+BDBR_{v}\right)}{8},$$
where ${BDBR}_{y}$, ${BDBR}_{u}$, and ${BDBR}_{v}$ are the weighted average of the BDBRs of the Y, U, and V components, respectively. Using ${BDBR}_{y}$, we compared the coding efficiency of the proposed and existing methods with respect to the anchor.

The encoding time reductions of the proposed and existing methods with respect to the anchor were assessed in each sequence. ΔEncT is calulated as: (4)$$\Delta EncT=\sqrt[{4}]{\prod_{QP_{i}\in 22,27,32,37}\frac{T_{method}\left(QP_{i}\right)}{T_{org}\left(QP_{i}\right)}},$$

To evaluate the model performances, we selected a method based on the traditional context-based approach for early TT partitioning [11]. The $BDBR_{y}$ and ΔEncT of the proposed method were evaluated at the α values of 0.5 and 0.25. For a fair comparison, VTM4.0 was applied to the existing and proposed methods. The experimental results are presented in Section 4.5.

### 4.3. Performance of ML Models for Accurate TT-Split Prediction

Table 4 and Table 5 present the performance results (accuracy and training time) of the ML models established for the TT_H and TT_V split directions, respectively.

The ML models used in the TT-split decision stage were DT, RF, and multi-layer perceptron (MLP) [9]. We first established three DT models with different maximum depths (max depth = 5, 6, and 7) and then established three RF models with the different number of DTs (number of DTs = 5, 6, and 7). We finally constructed a fully connected neural network with 13 input nodes, 30 hidden nodes, and 1 output node (the MLP model) and set the number of epochs to 2000 or 3000. The number of hidden layers was set to 30 to ensure the same accuracy for evaluating the proposed method as that for the existing method [11].

The results show that the DT models achieved higher accuracy within less training time than the other models. The DT model with max depth = 7 achieved the highest accuracy within a fast total training time in TT_H decisions. Thus, this model was selected for determining whether a TT-split is required in the TT-partitioning decision stage.

### 4.4. Performance of the Proposed Object-Cooperated TT Partitioning Decision Method

We now compare the performance of the method that inputs context-based features and the proposed method that additionally inputs object-features. Table 6 and Table 7 display the accuracy of the methods per sequence in the horizontal and vertical directions of TT-split, respectively, on a 0th frame of 22 sequences at various video resolutions [25].

Using the existing method, we evaluated a DT model with max depth 7 and only context-based features. In the proposed object-cooperated method, the TT-partitioning decision method, the DT model with a max depth = 7 was trained using 13 features comprising 11 context-based features and two additional features (object-features) obtained via object detection—employing YOLOv5.

As shown in Table 6, our proposed object-cooperated method exhibits higher accuracy than the DT-based method [9] in the worst cases (video sequences with an accuracy of less than 80%). In the worst cases, we also prove that our proposed method improves the accuracy of five out of seven sequences, as shown in Table 7, confirming its effectiveness.

### 4.5. Complexity-Reduction Performances of the MLP-Based and Proposed Method for Encoding-Complexity

Table 8 compares the performances of the existing and proposed method with respect to $BDBR_{y}$ and ΔEncT. To demonstrate that our proposed method is flexible for users depending on the need of applications, we adjusted the α value of classification on DT model for TT_H and TT_V. The ΔEncT value was optimized using the proposed method (with α=0.5). The best result of ΔEncT is the application of the proposed method when α=0.5 with a 60%, on average, compared with the anchor (VTM4.0). To list methods that show the superior performance based on ΔEncT, they are in the order of the proposed method (α=0.5) [11], and the proposed method (α=0.25). We also confirm that out proposed method reasonably reduced the encoding complexity of VVC. Meanwhile, the $BDBR_{Y}$ value when using the the proposed method (α=0.5) increased by 0.56%, which is 0.01% higher than that obtained using a previously reported model [11]. However, the value obtained using the proposed method (α=0.25) increased by only 0.11% relative to the anchor, although the ΔEncT value was 75%. Thus, our proposed method achieved a moderate trade-off between encoding complexity and coding efficiency.

The results of the video sequence experiments show that the proposed methods (α=0.5 and α=0.25) outperformed the method reported in [11] in terms of ΔEncT and $BDBR_{y}$, respectively. The largest reduction in encoding time was 57%, achieved using our proposed method with α=0.5 on the RaceHorses (832 × 480), Johnny sequence. On the same sequence, at the resolutions of (832 × 480) and (416 × 240), the existing MLP-based method reduced by 61% and 62%, respectively. Comparing the best results, it can be seen that our proposed method (α=0.5) improved by 4% and 5% in terms of ΔEncT, respectively, over the MLP-based method.

Table 9 shows results between the bitrate and the average object’s number, the object’s ratio when the DT model sets α as 0.5. The average object’s number and the object’s ratio were determined by object detection of frames of the JVET test sequences. As the result, we identified the assumption that object-features can be hints to determine the characteristics of the video. Based on various JVET test sequences [25], it was confirmed that sequences with a low object ratio or a small number of objects are superior to other sequences in terms of bitrate. For example, BQSquare and PartyScene sequences show a low average object ratio and the best bitrate. The MLP-based method [11] was incomparable because there were no object features.

Figure 6 and Figure 7 show the decoded images of models yielding the best ΔEncT results on the video sequence of RaceHorses (832 × 480) and RaceHorses (416 × 240) in Table 8 for QPs of 22 and 37, respectively. The image-quality degradations were not noticeably different in the proposed method, the MLP-based method [11], and VTM4.0. Meanwhile, Figure 8 and Figure 9 show the decoded images of models yielding the worst ΔEncT results in Table 8 for QPs of 22 and 37, respectively. On the video sequences of RitualDance and Cactus, where the proposed method (with α=0.5) delivered the poorest performance (68% and 71%, respectively), the encoding times were increased by 72% and 73%, respectively, in the existing method. Comparing the worst results, it can be seen that our proposed method (α=0.5) improved by 4% and 2% in terms of ΔEncT, respectively, over the MLP-based method. Moreover, increasing the QP from 22 to 37 caused no significant difference in the image-quality degradation of the proposed method, the MLP-based method [11], and VTM4.0.

## 5. Conclusions

To reduce the encoding complexity of VVC, we proposed a framework combining DL for object detection with ML for accurate TT-split prediction. The framework extracts the image features in the first stage and decides whether to split the TT in the second stage. In the feature extraction stage, we acquired object-features—object number and object ratio—using the object detection model YOLOv5. In the TT-split decision stage, we determined whether to split TT using DT, which showed the highest accuracy in an experimental test on multiple ML models. The experimental results confirmed that our proposed method is flexible for the purpose of the application. Therefore, the proposed method could be effectively used in the case of a VVC encoder that can reduce encoding complexity while somewhat compromising the quality or in the case of the encoder that can reduce encoding complexity while ensuring some quality. Furthermore, the extracted object-features and the optimization of the VVC encoder based on object detection may be further investigated to reduce the complexity of high-quality encoders in the near future.

## Figures and Tables

**Figure 1 sensors-22-06328-f001:**
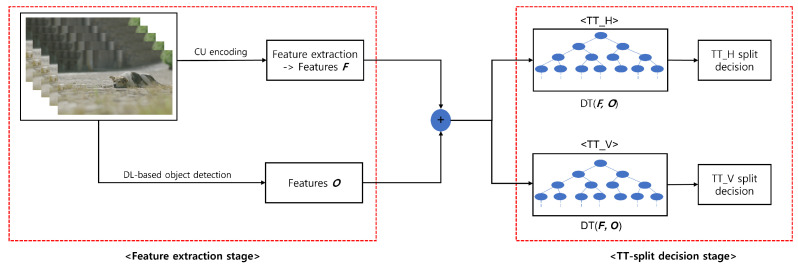
Framework of the proposed object-cooperated ternary tree (TT) partitioning decision method. Features *F* denotes the context-based features obtained by coding unit (CU) encoding, and Features *O* denotes the object-features obtained via object detection.

**Figure 2 sensors-22-06328-f002:**
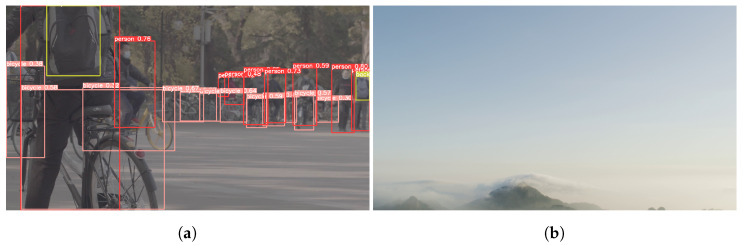
Results of YOLOv5-based object detection in two frames of video sequences: (**a**) most objects are detected; (**b**) no object is detected.

**Figure 3 sensors-22-06328-f003:**
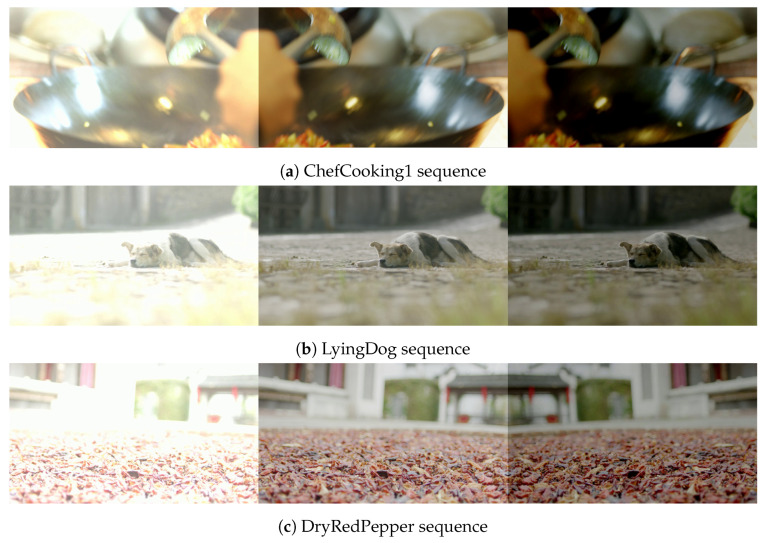
Examples of TVD sequences after data augmentation: a lightened image (**left**), the original image (**center**), and a darkened image (**right**).

**Figure 4 sensors-22-06328-f004:**
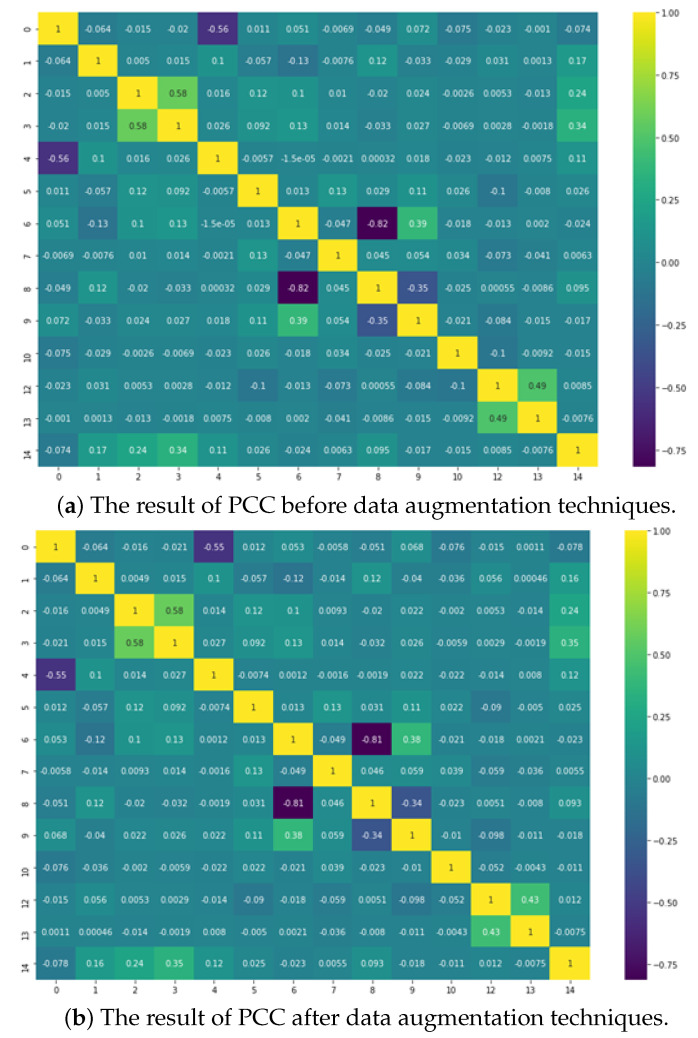
Heatmaps of the correlations between the features and binary class using the Pearson correlation coefficient (**a**) before and (**b**) after data augmentation.

**Figure 5 sensors-22-06328-f005:**
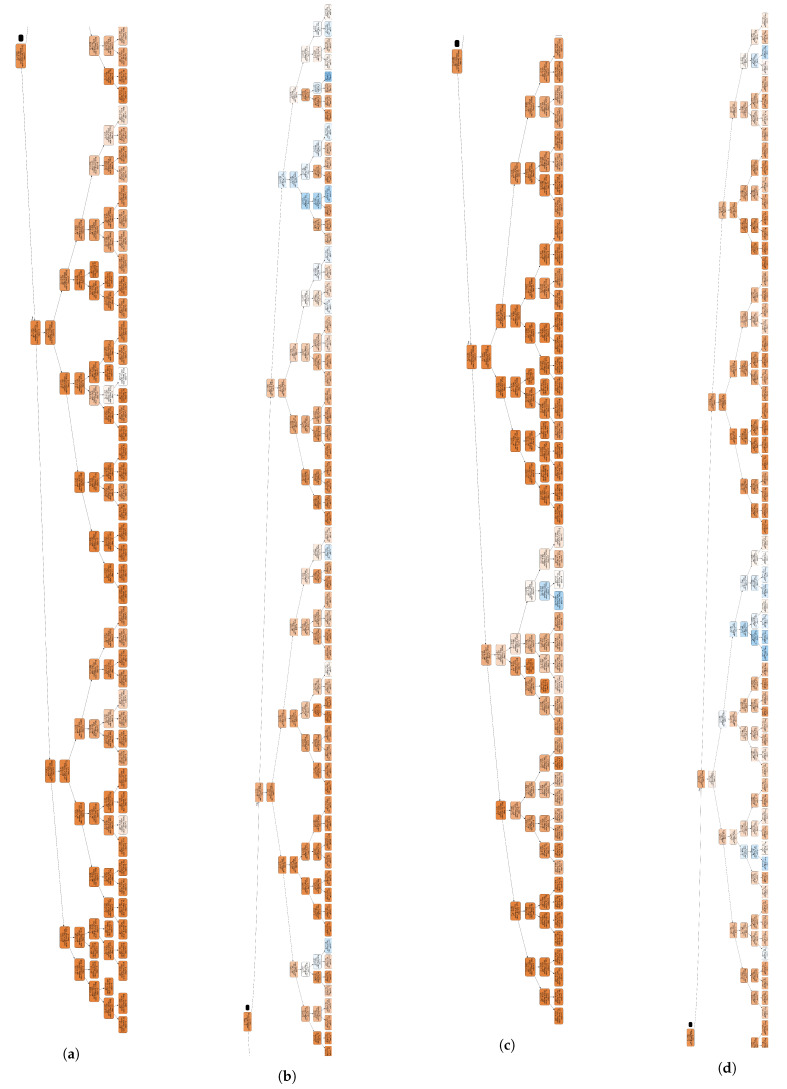
Graphs of decision tree (DT) models for (**a**,**b**) horizontal TT (TT _H), (**c**,**d**) vertial TT (TT _V) with a left and right by root node, respectively.

**Figure 6 sensors-22-06328-f006:**
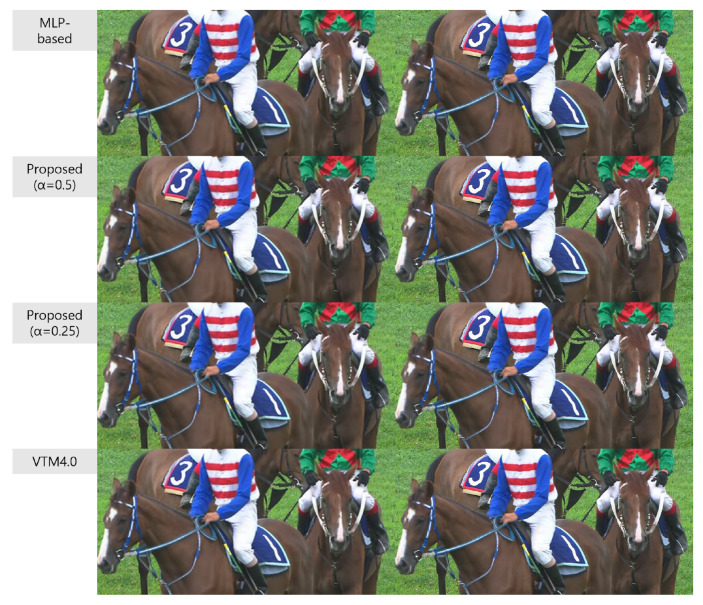
Decoded frames from RaceHorses (832 × 480) and RaceHorses (416 × 240) videos yielding the best ΔEncT results presented in Table 8 when QP = 22.

**Figure 7 sensors-22-06328-f007:**
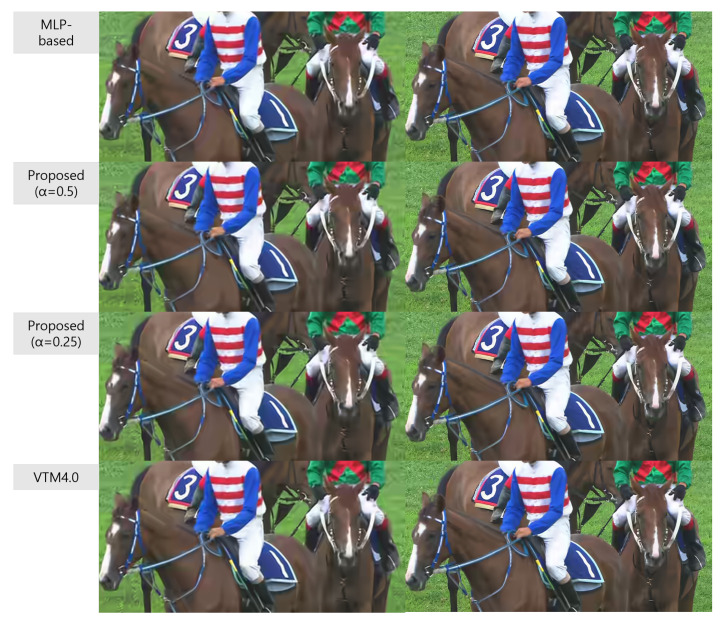
Decoded frames from RaceHorses (832 × 480) and RaceHorses (416 × 240) videos yielding the best ΔEncT results presented in Table 8 when QP = 37.

**Figure 8 sensors-22-06328-f008:**
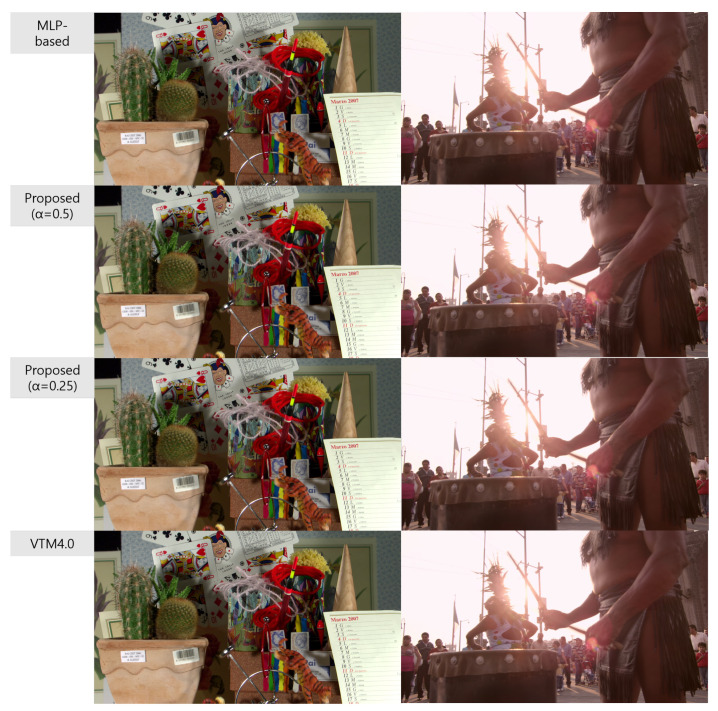
Decoded frames from Cactus and RitualDance videos yielding the worst ΔEncT results presented in Table 8 when QP = 22.

**Figure 9 sensors-22-06328-f009:**
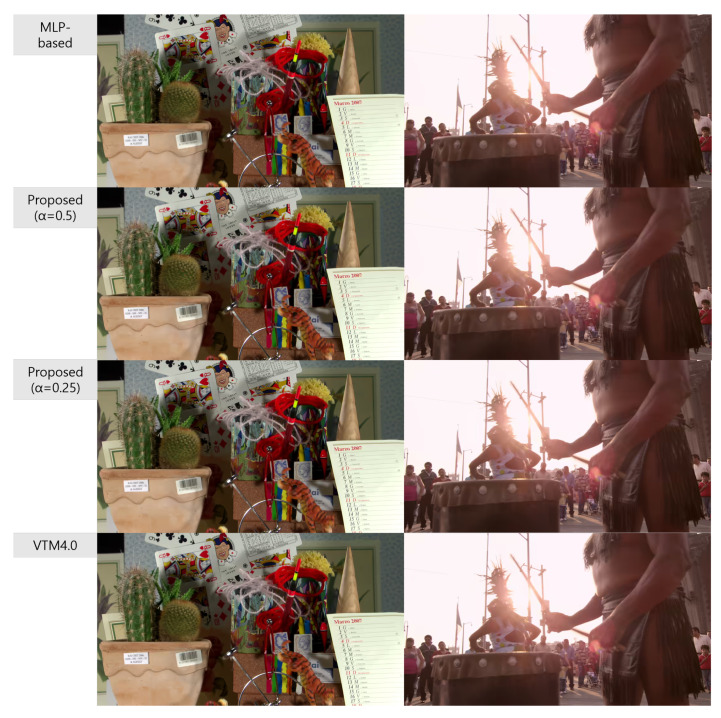
Decoded frames from Cactus and RitualDance videos yielding the worst ΔEncT results presented in Table 8 when QP = 37.

**Table 1 sensors-22-06328-t001:** Data augmentation by brightness adjustment of sequences.

Sequence	Brightness (%)
BoyDressing1	39.06
BoyDressing1	39.06
BoyMakingUp1	39.06
BoyMakingUp2	39.06
BoyWithCostume	39.06
ChefCooking1	39.06
ChefCooking2	39.06
ChefCooking3	39.06
ChefCooking4	39.06
ChefCooking5	39.06
ChefCutting1	−19.53
ChefCutting2	−19.53
DryRedPepper	−19.53
Fountain	−19.53
GirlWithTeaSet1	−19.53
GirlWithTeaSet2	−19.53
GirlWithTeaSet3	−19.53
HotelClerks	−19.53
HotPot	−19.53
LyingDog	−19.53

**Table 2 sensors-22-06328-t002:** The number of training samples for horizontal ternary tree (TT_H) and vertical ternary tree (TT_V) of sequences.

Sequence	TT_H	TT_V
Original	5,690,877	6,094,181
Brightness (39.06%)	302,072	304,357
Brightness (−19.53%)	672,066	290,886
Total	6,665,015	6,689,424

**Table 3 sensors-22-06328-t003:** Effectiveness of data augmentation techniques in the basic machine learning (ML) model with decision tree (DT) models (max_depth = maximum tree depth).

Model [9]	Accuracy (%) When Not Applied	Accuracy (%) When Applied
DT (max depth = 5)	86.66	86.66
DT (max depth = 6)	86.77	86.78
DT (max depth = 7)	86.79	86.85

**Table 4 sensors-22-06328-t004:** Performance results of the ML models for TT_H decisions.

Model [9]	Accuracy (%)	Model Training Time (s)
DT (max depth = 5)	86.66	32.1842
DT (max depth = 6)	86.78 ^1^	36.1361
DT (max depth = 7)	**86.85** ^1^	40.5430
RF (number of DTs = 5)	85.65	89.8891
RF (number of DTs = 6)	85.92	109.0825
RF (number of DTs = 7)	85.91	124.2268
MLP (epoch = 2000)	86.74	64,322.8483
MLP (epoch = 3000)	86.78 ^1^	92,569.8476

^1^ The most and second most accurate results are indicated in bold and blue fonts, respectively.

**Table 5 sensors-22-06328-t005:** Performance results of the ML models for TT_V decisions.

Model [9]	Accuracy (%)	Model Training Time (s)
DT (max depth = 5)	87.09	21.5739
DT (max depth = 6)	**87.17** ^1^	25.6598
DT (max depth = 7)	87.13 ^1^	27.7193
RF (number of DTs = 5)	85.37	57.9501
RF (number of DTs = 6)	85.79	68.3354
RF (number of DTs = 7)	85.61	80.5442
MLP (epoch = 2000)	86.97	60,731.3039
MLP (epoch = 3000)	87.03	91,888.3900

^1^ The most and second most accurate results are indicated in bold and blue fonts, respectively.

**Table 6 sensors-22-06328-t006:** Accuracy of the DT-based and proposed method regarding the TT_H decisions.

Sequence	DT-Based Method [9]	Proposed Method
	Accuracy (%)	Accuracy (%)
Tango2	**90.72** ^1^	90.71
FoodMarket4	96.84	96.84
Campfire	86.87	86.87
CatRobot1	**89.91** ^1^	89.87
DaylightRoad2	84.45	84.45
ParkRunning3	**90.08** ^1^	90.06
MarketPlace	87.09	**87.10** ^1^
RitualDance	85.63	**85.67** ^1^
Cactus	83.15	83.15
BasketballDrive	**81.24** ^1^	81.19
BQTerrace	81.29	**81.38** ^1^
BasketballDrill	**80.99** ^1^	80.96
BQMall	**81.54** ^1^	81.48
PartyScene	75.52	75.61 ^2^
RaceHOrsesC	80.97	**81.09** ^1^
BasketballPass	80.15	**80.22** ^1^
BQSquare	74.90	74.94 ^2^
BlowingBubbles	81.37	**81.46** ^1^
RaceHorses	82.62	82.65
FourPeoples	84.44	**84.49** ^1^
Johnny	88.32	**88.35** ^1^
KristenAndSara	86.87	**86.89** ^1^
Average	86.85	86.85

^1^ The most accurate results are indicated in bold. ^2^ If the accuracy is less than 80%, the more accurate of the two methods is displayed in blue.

**Table 7 sensors-22-06328-t007:** Accuracy of the DT-based and proposed method regarding the TT_V decisions.

Sequence	DT-Based Method [9]	Proposed Method
	Accuracy (%)	Accuracy (%)
Tango2	**90.78** ^1^	90.72
FoodMarket4	**96.69** ^1^	96.63
Campfire	83.54	**83.55** ^1^
CatRobot1	**85.58** ^1^	85.55
DaylightRoad2	**88.44** ^1^	88.43
ParkRunning3	90.54	**90.56** ^1^
MarketPlace	**90.83** ^1^	90.81
RitualDance	82.94	**82.97** ^1^
Cactus	85.78	85.78
BasketballDrive	**95.20** ^1^	95.19
BQTerrace	82.90	**82.98** ^1^
BasketballDrill	**81.65** ^1^	81.64
BQMall	77.58	77.68 ^2^
PartyScene	76.67	76.71 ^2^
RaceHOrsesC	82.48	82.48
BasketballPass	**82.28** ^1^	82.21
BQSquare	77.24 ^2^	77.20
BlowingBubbles	77.32	77.50 ^2^
RaceHorses	77.59	77.62 ^2^
FourPeoples	79.58	79.63 ^2^
Johnny	80.89	**80.98** ^1^
KristenAndSara	79.91 ^2^	79.90
Average	87.13	87.13

^1^ The most accurate results are indicated in bold. ^2^ If the accuracy is less than 80%, the more accurate of the two methods is displayed in blue.

**Table 8 sensors-22-06328-t008:** Comparison of the existing and proposed methods relative to the anchor (VTM 4.0).

Sequence	Resolution	MLP-Based Method [11]		Proposed Method		Proposed Method	
		α = 0.5		α = 0.5		α = 0.25	
		${\mathrm{BDBR}}_{Y}$	ΔEncT	${\mathrm{BDBR}}_{Y}$	ΔEncT	${\mathrm{BDBR}}_{Y}$	ΔEncT
MarketPlace	1920 × 1080	0.47%	66%	0.47%	64%	0.09%	87%
RitualDance	1920 × 1080	0.57%	72%	0.62%	68%	0.13%	89%
Cactus	1920 × 1080	0.48%	73%	0.52%	71%	0.07%	90%
BasketballDrive	1920 × 1080	0.73%	57%	0.56%	59%	0.12%	73%
BQTerrace	1920 × 1080	0.44%	61%	0.50%	58%	0.09%	73%
BasketballDrill	832 × 480	0.83%	61%	0.83%	59%	0.19%	72%
BQMall	832 × 480	0.50%	62%	0.51%	59%	0.07%	73%
PartyScene	832 × 480	0.32%	62%	0.30%	59%	0.04%	71%
RaceHorses	832 × 480	0.39%	61%	0.41%	**57%** ^1^	0.08%	74%
BasketballPass	416 × 240	0.51%	62%	0.45%	59%	0.17%	73%
BQSquare	416 × 240	0.25%	63%	0.27%	60%	0.02%	72%
BlowingBubbles	416 × 240	0.30%	62%	0.41%	58%	0.07%	71%
RaceHorses	416 × 240	0.25%	62%	0.32%	**57%** ^1^	0.00%	72%
FourPeople	1280 × 720	0.69%	62%	0.79%	58%	0.15%	74%
Johnny	1280 × 720	0.60%	61%	0.65%	**57%** ^1^	0.16%	72%
KristenAndSara	1280 × 720	0.53%	62%	0.59%	58%	0.11%	72%
Average		0.55%	63%	0.56%	60%	0.11%	75%

^1^ The best results in terms of ∆EncT are indicated in bold.

**Table 9 sensors-22-06328-t009:** Results between the bitrate and the average object’s number, object’s ratio when DT model sets α as 0.5.

Sequence	Resolution	Bitrate	Average of Object’s Number	Average of Object’s Ratio
MarketPlace	1920 × 1080	**0.47%** ^1^	9.77	0.38%
RitualDance	1920 × 1080	0.62%	18.57	0.54%
Cactus	1920 × 1080	**0.52%** ^1^	1.76	0.25%
BasketballDrive	1920 × 1080	0.56%	8.01	0.33%
BQTerrace	1920 × 1080	**0.50%** ^1^	22.42	0.10%
BasketballDrill	832 × 480	0.83%	7.15	0.29%
BQMall	832 × 480	**0.51%** ^1^	11.17	0.51%
PartyScene	832 × 480	**0.30%** ^1^	6.06	0.11%
RaceHorses	832 × 480	**0.41%** ^1^	4.71	0.80%
BasketballPass	416 × 240	**0.45%** ^1^	5.20	0.51%
BQSquare	416 × 240	**0.27%** ^1^	24.61	0.36%
BlowingBubbles	416 × 240	**0.41%** ^1^	3.85	0.88%
RaceHorses	416 × 240	**0.32%** ^1^	4.78	0.80%
FourPeople	1280 × 720	0.79%	18.69	0.58%
Johnny	1280 × 720	0.65%	12.56	0.63%
KristenAndSara	1280 × 720	0.59%	10.10	0.70%
Average		0.56%	10.58	0.48%

^1^ Superior results to average of bitrate are indicated in bold.

## Data Availability

Not applicable.

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
