# Peer review of "Object-Cooperated Ternary Tree Partitioning Decision Method for Versatile Video Coding"

_sensors, 2022, doi:10.3390/s22176328_

Round 1

Reviewer 1 Report

General Comments:

In this paper the authors propose a machine-learning based method to reduce the encoding complexity of the emerging versatile video coding (VVC) standard.

The topic is both timely and relevant. However, the paper has some weaknesses that must be solved in order to make it acceptable for this journal: First, the English writing must be carefully reviewed as there are many incomplete and unclear sentences that prevent an adequate understanding of the work presented (this exemplified in the detailed comments). Second, the technical description is not always clear, being sometimes rather vague and sometimes overly repetitive. Third, the experimental results are not sufficiently analyzed and do not always support the claims of the authors.

Detailed Comments:

Page 1:

  Content: "However, we assume that one of the important features is high-level objects that can provide hints on the characteristics of each video to decide whether complex TT partitioning is needed."

  Comment: This idea is not convincingly demonstrated in this paper.

Page 1:

  Content: "In particular, DL models such as object detection techniques have been successful to a variety of computer vision."

  Comment: Incomplete sentence.

Page 1:

  Content: "DL such as object detection techniques have rarely been deployed in video coding."

  Comment: Unclear sentence.

Page 1:

  Content: " However, few papers seem to have applied DL models such as object detection models to reduce video coding complexity of, particularly the next-generation video coding standard, versatile video coding (VVC)"

  Comment: This sentence can be misleading, as we have witness in the last years intensive research activities in applying DL methods to various aspects of the video coding/decoding processing chain.

Page 1:

  Content: "However, VVC introduced with the MTT structure (…)"

  Comment: Revise this sentence.

Page 1:

  Content: "because VVC encoder try to find"

  Comment: because VVC encoder tries to find

Page 2:

  Content: "are scares for fast TT partitioning"

  Comment: Check this sentence.

Page 3:

  Content: "Methods for object detection can be categorized in two:"

  Comment: Methods for object detection can be categorized in:

Page 3:

  Content: "historical of oriented gradients (HOG) functions"

  Comment: Check this acronym!

Page 3:

  Content: " COCO dataset."

  Comment: Provide a reference.

Page 3:

  Content: "We use YOLOv5s which has the fastest inference speed among YOLOv5 pre-trained models. 32."

  Comment: Check !

Page 3:

  Content: "VVC encoder try to find"

  Comment: VVC encoder tries to find

Page 3:

  Content: " because the encoder is rarely know"

  Comment: because the encoder rarely knows

Page 3:

  Content: "In feature extraction stage, we created new features"

  Comment: The authors repeat many times the term “new features” without explaining or giving some hints regarding these features.

Page 4:

  Content: "First, we extracted existing features (Feature F) through context-based approaches."

  Comment: Give examples of such features.

Page 4:

  Content: "As the features are listed in order from feature 0, they are quad-tree depth, BT’s superiority, BTD, block shape ratio, MTT depth, IPM, ISP, MRL, CBF, MTS, and quantization parameter."

  Comment: Acronyms should be defined the first time they are used.

Page 4:

  Content: " object ratio of the frame of video sequences. "

  Comment: What is object ratio ?

Page 5:

  Content: " For training dataset, the first, 21st, 41st, and 61st frames were used for all TVD sequences "

  Comment: Not clear which frames were used for training.

Page 5:

  Content: "Additionally, when we compare performance of existing and proposed methods for encoding complexity reduction, we used Feature Y obtained by object detection every 8 frames of each test video sequences."

  Comment: Why not using the same frames for both kind of tests ?

Page 6:

  Content: "When data augmentation techniques are applied, the accuracy of DT models is higher than when the techniques are not applied. Thus, we applied data augmentation techniques to training datasets."

  Comment: The authors must check if the presented values are percentages or not (it seems they are not).

Nevertheless, It can hardly be justified that, based on the results presented, data augmentation significantly produces different results. This needs to be verified and adequately commented.

Page 6:

  Content: "To predict the value of a target, we utilized gini impurity"

  Comment: Provide a reference.

Page 8:

  Content: "Figure 5 shows graphs of decision tree models when max depth is 7."

  Comment: The authors could exemplify the content of some blocks.

Page 10:

  Content: " The encoding time reduction (∆EncT) is as follows"

  Comment: Explain why using a geometric mean.

Page 11:

  Content: "A first frame of 22 sequences with various video resolution [23] was tested."

  Comment: How were these sequences selected? Why not using more frames ?

Page 11:

  Content: "Table 6. Accuracy of existing and proposed methods for TT_Hdecision."

  Comment: According to this table the two methods have equal average performance gains.

Page 12:

  Content: "Thus, we confirm that our proposed method achieve a moderate the trade-off between the encoding complexity and coding efficiency."

  Comment: Revise this sentence.

Page 16:

  Content: "The framework is consist of two-stage: feature extraction stage and TT split decision stage."

  Comment: Revise this sentence.

Page 16:

  Content: "In feature extraction stage, we gained new object-cooperated features, object number and object ratio, by object detection, YOLOv5."

  Comment: There are not sufficient evidence of the merits of this module, and the idea is not adequately explained.

Moreover, some more ablation studies are needed to infer the merits of the proposed method(s).

Author Response

We appreciate the reviewer for their constructive and valuable comments. During the stage of revision, we revised the original manuscript very carefully to accommodate all of the reviewers’ concerns. Since the comments contained multiple points, we divided each reviewer’s comments into sub-comments, and numbered them in order. In addition, modified texts are highlighted in the revised manuscript. Thank you again and we look forward to hearing from you soon.

Reviewer 2 Report

Well writtem, clear and easy to read.

Interesting subject matter.

Well presented and studied results

Specifics:

l24: le -> learning

l94: historical -> histogram

Figure 5 is near useful due to the size.

Figure 6/7/8/9 dont show much, for how much space they use

Author Response

(The authors gave the same response as above.)

Reviewer 3 Report

The work of this paper can be regarded as the optimization of literature [10]. Overall, this is a good job.

In Table 8, the author compares this work with literature [10]. But this comparison is not sufficient. This is because when [10] is applied, the alpha value is not specified. At the same time, this paper does not point out the performance of [10] when alpha =0.5/0.25.

Author Response

(The authors gave the same response as above.)
